# Impact of Pregnancy on Intra-Host Genetic Diversity of Influenza A Viruses in Hospitalised Women: A Retrospective Cohort Study

**DOI:** 10.3390/jcm8111974

**Published:** 2019-11-14

**Authors:** Gregory Destras, Maxime Pichon, Bruno Simon, Martine Valette, Vanessa Escuret, Pierre-Adrien Bolze, Gil Dubernard, Pascal Gaucherand, Bruno Lina, Laurence Josset

**Affiliations:** 1Virpath, INSERM U1111, CNRS UMR5308, International Center for Infectiology Research, ENS Lyon, Claude Bernard Lyon 1 University, 69008 Lyon, France; gregory.destras@chu-lyon.fr (G.D.); maxime.pichon@chu-poitiers.fr (M.P.); vanessa.escuret@chu-lyon.fr (V.E.); bruno.lina@chu-lyon.fr (B.L.); 2Virology Laboratory, Infectious Agents Institute, CBN, Groupement Hospitalier Nord, Hospices Civils de Lyon, 69004 Lyon, France; sib0.smb@gmail.com (B.S.); martine.valette@chu-lyon.fr (M.V.); 3Centre National des Virus des infections Respiratoires, Infectious Agents Institute, CBN, Groupement Hospitalier Nord, 69004 Lyon, France; 4Service de Chirurgie Gynécologique et Oncologique—Obstétrique, Centre Hospitalier Lyon Sud, Hospices Civils de Lyon, 69310 Pierre-Bénite, France; pierre-adrien.bolze@chu-lyon.fr; 5Hospices Civils de Lyon, Service de Gynécologie et d’Obstétrique, Hôpital de la Croix Rousse, 69004 Lyon, France; gil.dubernard@chu-lyon.fr; 6Consultation Obstétrique, Groupement Hospitalier Est, Hospices Civils de Lyon, 69500 Bron, France; pascal.gaucherand@chu-lyon.fr

**Keywords:** influenza, pregnancy, quasispecies, ultra-deep sequencing, viral diversity

## Abstract

Characterising dynamics of Influenza A Viruses (IAV) within-host evolution is an active field of research which may lead to a better understanding of viral pathogenesis. Using a pregnant mouse model, a study has recently suggested that immune modulation during pregnancy could promote the emergence of IAV quasispecies with increased virulence. Herein, we assess the clinical relevance of these findings in humans. We studied IAV intra-host diversity (ihD) in pregnant (*n* = 36) and non-pregnant (*n* = 23) women hospitalized in Lyon for IAV infection (01/2015–05/2018). Whole IAV genomes present in nasopharyngeal samples were sequenced in duplicate to analyze reproducible intra-host single nucleotide variants (ihSNV). Counts, relative frequencies and locations of ihSNV were used as indicators of ihD. The median ihSNV/kb counts per segment were between 0 and 1.3. There was >81% ihSNV at relative frequencies between 1–5% for H1N1 and >51% for H3N2 IAV. No significant difference was noted between pregnant and non-pregnant women when considering all or only non-synonymous ihSNV. Seven convergent non-synonymous ihSNV were found; none were significantly associated with pregnancy. These results suggest that modulation of the immune system during pregnancy in humans does not impact IAV ihD, in contrast to mice.

## 1. Introduction

Due to its error-prone polymerase, RNA eight-segmented influenza A viruses (IAV) acquire genetic diversity when replicating [1]. Understanding intra-host diversity (ihD) of IAV is of great interest because ihD may shape IAV evolution at the inter-host scale but also impact IAV pathogenesis [2]. Several factors, including host immune competence, might modulate ihD. Recently, using allogeneic pregnant mice infected with human A(H1N1)pdm09, Engels et al. suggested that immune modifications of both innate and adaptive immunity during pregnancy could drive the emergence of convergent non-synonymous mutations (Q223R on HA according to HA1 numbering, R211K on NS1, and R54N on NEP) associated with fatal outcomes [3]. They postulated that this mechanism could explain why pregnant women are more at risk of developing severe influenza. To examine the clinical relevance of these findings, we assessed IAV ihD in women according to pregnancy status and its potential impact on fostering severe forms.

## 2. Materials and Methods

### 2.1. Study Cohort

All women aged 15–45 years admitted to the emergency departments of the Lyon university hospital (France) with nasopharyngeal swabs positive for IAV during three consecutive influenza seasons (01/2015–05/2018) were retrospectively identified. Women with missing clinical record and/or samples were excluded. Patients were classified according to pregnancy status and influenza severity (defined as the need for mechanical ventilation and/or cardiac complication—myocarditis or pericarditis). 

### 2.2. IAV Whole Genome Amplification and Sequencing

IAV whole genome amplification was performed directly on IAV-positive samples as previously described [4]. Briefly, an automatic extraction platform (Nuclisens EasyMag, Durham, NC, USA) was used, followed by a DNase treatment (TurboDNAse, LifeTechnologies, Carlsbad, California, CA, USA) before whole genome amplification by universal primers (Uni12 and Uni13). Sequencing was performed on a NextSeq 500 2 × 150 bp reads (Illumina, San Diego, California, US). Importantly, all respiratory samples were extracted and sequenced in duplicate to improve variant detection accuracy. This, however, required a high volume of samples (400 µL).

### 2.3. Bioinformatic Analysis

Reads were analyzed as previously described including i) quality trimming and filtering (>Q30) and removal of human reads; ii) mapping on genome references (A/pH1N1/California/07/2009 and A/H3N2/Perth/16/2009) to generate consensus sequences; and iii) intra-host Single Nucleotide Variants (ihSNV) calling using naive variant caller [5]. Genetic diversity was then analyzed using R [6] using the following criteria for each nucleotide position: depth >1000X; ihSNV detected in both duplicates with mean relative frequency >1%. Count and relative frequency of ihSNV were normalized by the number of analyzed positions (>1000X). 

### 2.4. Statistical Analysis

Statistical analyses were performed using R 3.5.3 (available at https://cran.r-project.org/bin/windows/base/old/3.5.3/). For qualitative variables, χ^2^ or G-test was used. For quantitative variables, the t-test or Mann-Whitney test was used as appropriate. Z-scores for ihSNV counts/kb were computed per segment for severe cases. A *p*-value < 0.05 was considered significant.

### 2.5. Ethical Statement

The study was approved by the local ethics committee of the Lyon University Hospital on 21/12/2015.

## 3. Results

In total, 284 women were identified; after exclusion of those with missing clinical records (*n* = 12) or insufficient volumes of samples (*n* = 187), and after exclusion of sequences with low coverage (*n* = 26), whole genome sequences of 59 IAV (36 pregnant and 23 non-pregnant women) were successfully obtained in duplicate. Pregnant women were significantly younger; there was no significant difference in the frequency of patients with an underlying condition, time since symptom onset, antiviral treatment before sampling, hospitalization for influenza, influenza severity, viral load or viral subtypes between pregnant and non-pregnant women (Table 1).

Regarding sequencing results, no significant difference of depth of coverage was found between pregnant and non-pregnant women (Appendix A). There was no association between phylogeny and pregnancy or clinical severity (Appendix A).

For diversity, the median ihSNV/kb counts per segment ranged from 0 (for four segments of H1N1 viruses and six segments of H3N2 viruses, in both pregnant and non-pregnant women) to 1.3 for the 5th segment (NP) of H3N2 viruses (Figure 1A,B). There was no significant difference in the number of ihSNV/kb according to pregnancy (Figure 1A,B); this was also found when only non-synonymous ihSNV/kb were considered (Appendix A). Regarding severe cases, all ihSNV z-scores computed for the different segments were lower than 2.6, both for pregnant women (range: −0.95 to 2.56 for H1N1 and −0.57 to 2.53 for H3N2) and non-pregnant women (range: −1.4 to 1.4 for H1N1 and −1.3 to 2.5 for H3N2).

For H1N1 viruses, 85% of the ihSNV were found at a low relative frequency (1–5%) in pregnant women and 81% in non-pregnant women (Figure 1C,D). H3N2 viruses were slightly more diverse; 65% of the ihSNV were found at a low relative frequency in pregnant women and 51% in non-pregnant women (Figure 1C,D). Distribution of ihSNV relative frequencies were not significantly different between pregnant and non-pregnant women (*p* = 0.16 for H1N1; *p* = 0.35 for H3N2). High-relative frequency ihSNV (>30%) were all synonymous, except for a G11S mutation in NA that was present in one pregnant woman with severe H1N1 influenza (Figure 1C,D, Appendix A). 

In total, 85 non-synonymous ihSNV were detected and dispersed across the viral genome (Appendix A). None of the mutations were associated with antiviral resistance. Only seven non-synonymous ihSNV (at a relative frequency between 1–10.5%) were shared between women (Table 2). Two highly prevalent non-synonymous ihSNV (range: 9–13 patients, irrespective of pregnancy) were detected on PA (E493A) and NP (T130K) of H1N1. Five low prevalent non-synonymous ihSNV (range: 3–5 patients, irrespective of pregnancy) were detected on H1N1 HA (P182Q, T342A, E356K, HA1 numbering) and on PB1 of both H3N2 and H1N1 (K279Q, Q460K). The K279Q and Q460K substitutions on H3N2 were found exclusively in three pregnant women, while H1N1 PB1 K279Q was found in four pregnant and one non-pregnant women. However, there was no significant difference in the prevalence of these mutations or their relative frequencies according to pregnancy (Table 2).

## 4. Discussion

In this cohort of 59 patients, we found no evidence of a specific impact of pregnancy on IAV ihD in humans. Most of the IAV infecting both pregnant and non-pregnant hospitalized women had low ihD, with less than 1.3 ihSNV/kb per segment and relative frequencies between 1–5%. None of the mutations previously found in the murine pregnancy model [3] were found, and no specific evolution of IAV during pregnancy was observed.

IAV ihD in both pregnant and non-pregnant women was very limited, which is in line with other studies showing that acute infections in immunocompetent patients are associated with less than 10 ihSNV per sample, most at relative frequencies 1–10% [7,8,9,10,11]. In these patients, vaccination was not associated with increased ihD or emergence of antigenic variants [8,10]. In contrast, immunosuppressed patients may develop chronic IAV infection leading to increased ihD, with up to 60 ihSNV per sample at relative frequencies >5%, including variants involved in antiviral resistance during treatment [12]. Pregnancy has been identified as a risk factor for hospitalization with influenza and consequently, vaccination of pregnant women and antiviral treatments are recommended. Our results suggest that immune modulation during pregnancy does not give rise to increased viral diversity. Therefore, infections in pregnant women are unlikely to enable the emergence of antigenic variants or resistance mutations, similarly to infections in immunocompetent patients.

This retrospective study does, however, have limitations. For instance, as only hospitalised patients were included, demographics and clinical data are not representative of the general population. In addition, only a few cases with severe influenza were included (two pregnant, five non-pregnant), limiting ihD analysis according to severity. Furthermore, only one nasopharyngeal sample/patient was sequenced at admission, while it would be interesting to compare the parallel evolution of ihD between pregnant and non-pregnant women, and between upper and lower respiratory tract. A strength of the study was that sequencing was performed in duplicate, increasing the validity of the variants identified, although very stringent criteria for ihD investigation leaded to exclusion of many samples. This is usually not done in studies of IAV quasispecies that suffer of low reproducibility for low viral loads [2].

Taken together, the results indicate that pregnancy is not associated with increased IAV ihD in humans. Although the only two pregnant women with severe influenza herein had very low IAV ihD, similar to non-pregnant women, we cannot exclude that IAV ihD may be increased in women with specific genetic or immune predisposition, but also with severe influenza. Nonetheless, the data presented herein suggest that the convergent mutations associated with severity in the pregnant mouse model (Q223R on HA, R211K on NS1, and R54N on NEP) [3] are not relevant in pregnant women. As these specific mutations were also observed in non-pregnant mice, although at a lower relative frequency, they might be associated with virus adaptation to mice rather than to pregnancy itself, as exemplified by Q223R. Indeed, Q223R mutation affects the receptor binding site affinity, and such mutation likely reflects IAV adaptation after inter-species infection [13,14]. 

Seven non-synonymous ihSNV on PA (E493A), NP (T130K) and HA (P182Q, T342A, E356K) of H1N1, and on PB1 (K279Q, Q460K) of both H3N2 and H1N1 viruses were shared by several samples. None has been reported elsewhere except for HA P182Q, detected in out- and hospitalised patients in Brazil during the 2009 pandemic [15]. None of these shared ihSNV were significantly associated with pregnancy but there was a trend towards higher occurrence of PB1 K279Q and Q460K in pregnant women. These results warrant further investigation in larger cohorts and experimental studies to assess the impact of these mutations on viral pathogenesis. Finally, while most of the ihSNV were found at different positions, we cannot exclude that some ihSNV might have similar effects on influenza virus fitness or virulence and that functional convergence might occur during influenza within-host evolution. 

## 5. Conclusions

This study suggests that pregnancy does not impact IAV ihD in humans, in contrast to mice. Further studies are needed to investigate whether influenza severity in humans is associated with specific IAV within-host evolution in the lung.

## Figures and Tables

**Figure 1 jcm-08-01974-f001:**
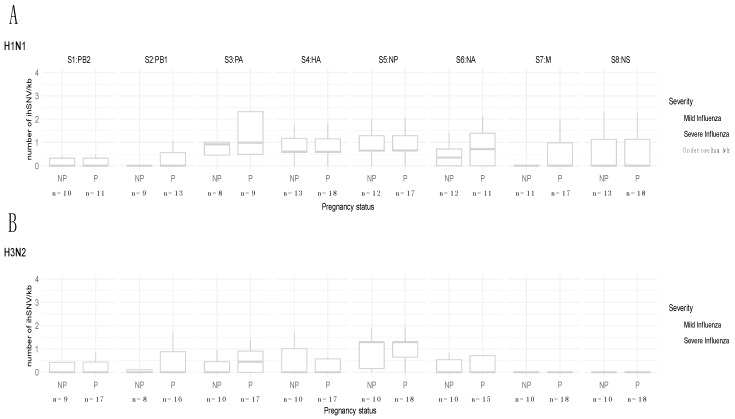
Comparison of Influenza A Viruses intra-host diversity (IAV ihD) between pregnant and non-pregnant women. Number of ihSNV/kb per segment (S1 to S8) in each sample, according to pregnancy status (P: pregnant; NP: non-pregnant) for H1N1 (**A**) and H3N2 (**B**) viruses. Each point represents a patient, with black dot for mild influenza and red triangles for severe influenza. Distribution of ihSNV relative frequencies according to pregnancy status (P: pregnant; NP: non-pregnant) for H1N1 (**C**) and H3N2 (**D**) viruses. Severe influenza is in red. One patient treated with oseltamivir is in green.

**Table 1 jcm-08-01974-t001:** Characteristics of women included in the study.

	Pregnant Women *n* = 36	Non-Pregnant Women *n* = 23	*p*
Age, mean years (SD)	30.6 (0.96)	34.7 (1.5)	0.03
Trimesters, *n* (%)			
Trimester 1	2 (5.6)		
Trimester 2	9 (25.0)		
Trimester 3	25 (69.4		
With underlying condition, *n* (%)	10 (30.3)	10 (43.5)	0.21
Time since onset of symptoms, median days (IQR)	2.0 (1.0–3.0)	2.0 (1.0–3.0)	0.80
Oseltamivir treatment before sampling, *n* (%)	1 (2.8)	0 (0.0)	1.00
Hospitalisation for influenza, *n* (%)	11 (30.6)	11 (47.8)	0.18
Severe influenza, *n* (%)	2 (5.6)	5 (21.7)	0.10
IAV viral load, mean Ct (SD)	26.01 (4.2)	26.35 (4.2)	0.80
IAV sub-type and clades, *n* (%)			
H1N1 6B1	18 (50)	13 (56.5)	0.96
H3N2	18 (50)	10 (43.5)	0.82
H3N2 3c2a1	13 (36.1)	9 (39.1)	0.65
H3N2 3c2a2	3 (8.3)	0 (0.0)	0.40
H3N2 3c2a3	2 (5.5)	1 (4.3)	0.83
H3N2 3c3b	1 (2.8)	0 (0.0)	1.00

SD: Standard deviation; IQR: interquartile range; *p*: *p*-value.

**Table 2 jcm-08-01974-t002:** Convergent non-synonymous ihSNV.

	Pregnant Women *n* = 36	Non-Pregnant Women *n* = 23	*p* Occurrence of NS-ihSNV	*p* Relative Frequencies
H1N1, women ratio* (min–max of ihSNV relative frequencies)				
S2-PB1				
K279Q	4/6 (2.3–8.6%)	1/6 (1.8%)	0.24	
Q460K	1/6 (1.6%)	0/6	1.00	
S3-PA				
E493A	9/9 (1.3–14.3%)	8/8 (1.7–10.5%)	1.00	0.55
S4-HA**				
P182Q	1/16 (4.6%)	4/12 (1.3–3.6%)	0.13	
T342A	2/13 (1.2–1.4%)	2/8 (1.2–1.7%)	0.53	>0.999
E356K	9/15 (1.2–7.6%)	3/12 (1.2–6.2%)	0.12	>0.999
S5-NP				
T130K	13/17 (1.1–1.8%)	9/12 (1.1–3.5%)	>0.99	0.07
H3N2, women ratio* (min–max of ihSNV relative frequencies)				
S2-PB1				
K279Q	4/13 (1.5–3.4%)	0/6	0.25	
Q460K	3/13 (1.3–2.0%)	0/6	0.52	

* Number of women with the same ihSNV on total number of women that were analysed at that position; ** HA1 numbering; NS-ihSNV: Non-Synonymous intra-host diversity.

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
