# Peer review of "Impact of Pregnancy on Intra-Host Genetic Diversity of Influenza A Viruses in Hospitalised Women: A Retrospective Cohort Study"

_jcm, 2019, doi:10.3390/jcm8111974_

Round 1

Reviewer 1 Report

Impact of pregnancy on intra-host genetic diversity of influenza A viruses in hospitalised women: a retrospective cohort study

Gregory Destras et al.

In this study, Destras et al describe the genetic diversity of H1N1pdm2009 and H3N2 influenza viruses during hospitalization of pregnant and non-pregnant women.

The authors do not find any particular differences in genetic diversity when comparing pregnant and non-pregnant women, as was previously suggested in a mouse study.

The approach used suits the research question and the results are properly discussed in a brief, compact report.

However, certain topics can be discussed in a bit more detail or could use some extra clarification as mentioned below.

Minor remarks:

Line 30-31: this hypothesis does not explain the observed differences between pregnant and non-pregnant mice and is therefore not suitable. Line 38: “Several factors, including host immune competence, might modulate ihD”. This might be discussed in a bit more detail to show evidence and implications. Line 41: which numbering is used for the HA residues throughout the paper? H3N2? With our without signal peptide? Line 47: study cohort. Did the patients receive any kind of treatment like Oseltamivir? This should be included in Table 1. Table headings should appear above tables, not underneath. Line 72-74: Sequences were obtained form only 59/284. Can the authors comment on rather low success rate? Is there any correlation between viral load (Ct value) or severity? Table 1: Please use range instead of standard deviation or IQR when describing medial meta data. Line 81: and further, Fig 1: It seems that many samples are missing in the graphs. There are not 59 data points for every segment? This is not clear from the coverage data? Fig 1: Are there any significant differences in genetic diversity between segments? Table S1: Is this total number of reads or average per sample? If total it is not very high for so many samples and the used cot-offs (>1000 reads) Line 88 and further, Fig 1C-D: The separation between pregnant and non-pregnant is confusing. The authors should consider using separate bars for the pregnant and non-pregnant samples, like they did split them in panel A and B. In case of OST treatment, did the authors observe any resistance associated mutations. Line 95: “Only 7 non-synonymous ihSNV (at a relative frequency between 1-10.5%) were shared” OF how many non-syn mutations in total? Line 100-102. The authors mention substitutions “exclusively” found in pregnant women, however Table S2 suggests otherwise? This section si therefore not completely clear. Line 95-102: This section could use a little more explanation/detail. How many non-syn mutations are found in total (not only the shared ones). What viral proteins, which domains do they appear. IS anything know about functional effects of the substitutions found or the domains they appear in? Fig S1: Any trend observed for other genes? The authors claim full genome sequencing, so phylogenetic trees of other segments can provide extra insights. Line 119: Other references can be included and discussed (e.g. Dinis & Friedrich JVI 2016, Xue & Bloom, Nat Gen 2019) Line 131: “While the only 2 pregnant women with severe influenza herein had very low IAV ihD” Like in non-pregnant women, right? Line 137 and further: like mentioned above, can the authors speculate on the function of the observed substitutions and whether alternative substitutions observed in their patients (maybe only once, or in multiple patients) could have similar effects despite being alternative positions?

Reviewer 2 Report

This manuscript addresses the very important question whether the decreased immune competence associated with pregnancy gives rise to increased viral diversity. Therefore, virus sequences from human nasopharyngeal swabs were retrospectively analyzed. The results indicate that intra host diversity is not increased during pregnancy.

The study is well performed and the results are technically correct analyzed and presented. However, the description of the results in the text could be improved. The interpretation of the data, their biological relevance and social impact is, in my opinion, insufficient and needs to be extended.

Although several mutations, such as K279Q and Q460K in PB1 are not “significantly” enriched according to p-values, there is a clear trend of their occurrence in pregnant women in comparison to non-pregnant women in this study. This trend needs to be discussed and acknowledged.

In addition, mutations K279Q and Q460K occurred in both strains H1N1 and H3N2 with increased prevalence in pregnant women, which speaks for an important biological role of these sites. Unfortunately, any mechanistic data on the biological relevance of these mutations for viral pathogenesis are not provided. Optimally, the impact of each mutation in the viral pathogenesis should be investigated in the context of recombinant viruses. However, these aspects need to be discussed on a broader scale as this is highly relevant for the risk of emergence of new virus variants with increased pathogenicity. In addition, the social impact of the study results should be discussed. What does this imply for vaccination and antiviral treatments for pregnant women.

Critical remarks on manuscript structure and content:

The Manuscript contains many incomplete, unfinished sentences and word duplications. Please check carefully.

Table legends are provided below the tables, please put on top of the table.

Also there is some unclear numbering of the figures and text (see Table 1, which is headed with “3.2 Figures, Tables and Schemes” this should be omitted.

Is the experimental section the material and methods part? This seems unusual. Please change the name of this section and check the journal web page for instructions on manuscript sections and structure.

Please describe the used methods in a detailed manner and avoid statements like “as previously described” (line 59).

Size and quality of Figure 1A) needs to be increased.

Please clarify the content of the sentence in line 92-94. Does the G11S mutation occur in 45% of H1N1 infected women or just in one pregnant women who developed a severe influenza?

The identity of the observed mutations is of high importance and should be considered as primary data in the manuscript. Therefore Table S2 should not be considered as supplementary but changed to table 2.

The conclusion in the first sentence of the Discussion (lines 113-114) is very strong. Given the rather small number of samples analyzed and the limited time frame of sample collection in this study, I strongly encourage the authors to carefully rephrase the conclusion in order to avoid an over-interpretation of the presented data.

Please explain what is meant by the term “mouse-associated mutations” in the context of this study.

Please avoid terms such as “low”, few and “high”, “increased” without referring to the actual numbers (lines 115 and 118, 120).
